# Vitamin D-Dimer: A Possible Biomolecule Modulator in Cytotoxic and Phagocytosis Processes?

**DOI:** 10.3390/biomedicines10081785

**Published:** 2022-07-25

**Authors:** Ralf Herwig, Katharina Erlbacher, Amela Ibrahimagic, Mehtap Kacar, Naime Brajshori, Petrit Beqiri, Joachim Greilberger

**Affiliations:** 1Laboratories PD Dr. R. Herwig, 80337 Munich, Germany; labor@pdherwig.de (R.H.); katharina.erlbacher@gmx.at (K.E.); 2Heimerer-College, 10000 Pristina, Kosovo; naime.brajshori@kolegji-heimerer.eu (N.B.); petrit.beqiri@kolegji-heimerer.eu (P.B.); 3Department of Speech and Language Pathology and Audiology, Faculty of Education and Rehabilitation, University of Tuzla, 75000 Tuzla, Bosnia and Herzegovina; amelaibr@gmail.com; 4Department of Physiology, Faculty of Medicine, Yeditepe University, Ataşehir, 34755 İstanbul, Turkey; mehtapkacar69@gmail.com; 5Department of Pathophysiology, Health Sciences Institute, Yeditepe University, Ataşehir, 34755 İstanbul, Turkey; 6Institut fuer Laborwissenschaften, 8301 Lassnitzhoehe, Austria; 7Division of Medicinal Chemistry, Otto-Loewi Research Center for Vascular Biology, Immunology and Inflammation, Medical University of Graz, 8010 Graz, Austria

**Keywords:** vitamin D_3_ (VitD_3_, cholecalciferol), vitamin D binding protein (VDBP), deglycosylated vitamin D binding protein (dgVDBP; GcMaf), vitamin D_3_ complexed to deglycosylated vitamin D binding protein (VitD-dgVDBP), calcidiol (calcifediol, 25OHD, 25-hydroxy-vitamin D), calcitriol (1*α*,25-dihydroxycholecalciferol, 1,25(OH)2D, or vitamin D hormone), reactive oxygen radicals (ROS), human peripheral blood mononuclear cell (PBMC), lipopolysaccharide (LPO)

## Abstract

Background: Vitamin D_3_ complexed to deglycosylated vitamin D binding protein (VitD-dgVDBP) is a water-soluble vitamin D dimeric compound (VitD-dgVDBP). It is not clear how VitD-dgVDBP affects circulating monocytes, macrophages, other immune cell systems, including phagocytosis and apoptosis, and the generation of reactive oxygen species (ROS) compared to dgVDBP. Methods: Flow cytometry was used to measure superoxide anion radical (O_2_^*−^) levels and macrophage activity in the presence of VitD-dgVDBP or dgVDBP. VitD-dgVDBP was incubated with normal human lymphocytes (nPBMCs), and several clusters of determination (CDs) were estimated. dgVDBP and VitD-dgVDBP apoptosis was estimated on malignant prostatic cells. Results: The macrophage activity was 2.8-fold higher using VitD-dgVDBP (19.8·10^6^ counts) compared to dgVDBP (7.0·10^6^ counts), but O_2_^*−^ production was 1.8-fold lower in favor of VitD-dgVDBP (355·10^3^ counts) compared to dgVDBP (630·10^6^ counts). The calculated ratio of the radical/macrophage activity was 5-fold lower compared to that of dgVDBP. Only VitD-dgVDBP activated caspase-3 (8%), caspase-9 (13%), and cytochrome-C (11%) on prostatic cancer cells. PE-Cy7-labeled VitD-dgVDBP was found to bind to cytotoxic suppressor cells, monocytes/macrophages, dendritic and natural killer cells (CD8+), and helper cells (CD4+). After 12 h of co-incubation of nPBMCs with VitD-dgVDBP, significant activation and expression were measured for CD16++/CD16 (0.6 ± 0.1% vs. 0.4 ± 0.1%, *p* < 0.05), CD45k^+^ (96.0 ± 6.0% vs. 84.7 ± 9.5%, *p* < 0.05), CD85k^+^ (24.3 ± 13.2% vs. 3.8 ± 3.2%, *p* < 0.05), and CD85k^+^/CD123^+^ (46.8 ± 8.1% vs. 3.5 ± 3.7%, *p* < 0.001) compared to the control experiment. No significant difference was found using CD3+, CD4+, CD8+, CD4/CD8, CD4/CD8, CD16+, CD16++, CD14+, or CD123+. A significant decline in CD14+/CD16+ was obtained in the presence of VitD-dgVDBP (0.7 ± 0.2% vs. 3.1 ± 1.7%; *p* < 0.01). Conclusion: The newly developed water-soluble VitD_3_ form VitD-dgVDBP affected cytotoxic suppressor cells by activating the low radical-dependent CD16 pathway and seemed to induce apoptosis in malignant prostatic cells.

## 1. Introduction

Vitamin D is involved in several processes of the adaptive and innate immune systems [1,2]. The innate immune system’s first-line defence against infections is derived from invading pathogens in which activated macrophages and monocytes express CYP27B1, which converts vitamin D to the more activated form calcitriol, alongside Toll-like receptor signalling and exposure to inflammatory cytokines, such as interferon-γ. Along an autocrine pathway, calcitriol stimulates antimicrobial activities via the vitamin D receptor pathway (VDR-RXR) in the generation of substances such as cathelicidin or defensin to destabilize microbial membranes and viral envelopes [3]. The conversion of vitamin D to calcitriol also takes place in dendritic cells during the expression of CYP27B1, which results in a downregulation of the major histocompatibility complex II (MHC II), antigen presentation, interleukin 12, and clusters of determination 40, 80, and 86 (CD40, CD80, CD86). This modulation of calcitriol on antigen-presenting cells, the upregulation of T-helper type 2 cells, and the generation of interleukins 4 and 10 inhibit inflammation and are accompanied by the downregulation of T-helper type 1 cells (Th1), T-helper type 17 cells (Th17), and T regulatory cells (Treg) in lowering inflammation caused by interferon γ, interleukins 2, 6, and 17, and tumoral necrosis factor alpha (TNFα) [4,5,6,7,8,9]. APC and Treg themselves induce apoptosis in autoreactive cells, playing a crucial role in the self-tolergenic pathway.

Cholecalciferol, vitamin D_3_, is mostly transported by vitamin D binding protein (VDBP), which is involved in the activation of phagocytosis in phagocytosing cells, such as monocyte-derived macrophages, as well as in the elimination of actin and fatty acids in necrotic lesions [10]. Due to the combination of a high affinity for vitamin D metabolites and a high protein concentration, free concentrations of all vitamin D metabolites are extremely low [11]. A question arose regarding the uptake of vitamin D into cells, which is assumed to occur by pinocytosis [12].

Vitamin D binding protein, as well as its close family members, is mainly produced in the liver, although the gene and protein are also expressed in very low concentrations in other tissues. The half-life of DBP in human plasma is about 1.7 days, and thus, markedly shorter than the half-life of 25OHD, which is estimated at about 15 days, based on studies with deuterium-labelled 25OHD in healthy subjects from the U.K. and Gambia [13]. VDBP has no direct effects on inflammation. Its ligand, 1,25(OH)2D, has many immune and inflammatory effects, as reviewed in [14,15], and VDBP inhibits the cellular entry of 1,25(OH)2D.

Vitamin D and its metabolites play important roles in the renal proximal tubule function bound to VDBP. Through a combination of studies on animals, molecular and cellular biologic studies, the use of knockout mice, and clinical observations in patients with Mendelian disorders and various forms of rickets, the interactions of vitamin D and proximal tubule function can be revealed. Filtered 25(OH)D bound to DBP is taken up by an endocytic process involving megalin and cubilin found in the brush border surface of the proximal tubule [16].

Several in vitro studies have suggested that the deglycosylation of DBP can activate DBP to become a macrophage-activating factor (DBP-MAF; dgVDBP). A membrane-bound β-galactosidase present on the surface of immune B cells and a sialidase present on T cells removed two of the three sugar residues of DBP/GC1 protein [17]. This DBP was then able to facilitate the differentiation of monocytes into osteoclasts and even corrected the osteoporosis phenotype of mice [18,19,20]. In other circumstances, DBP-MAF can activate macrophages in their battle against cancer cells or infections.

We have recently shown that VitD-dgVDBP, a newly developed water-soluble VitD_3_ form, has a high potential to activate macrophage phagocytosis compared to VDBP or T- and B-lymphocyte-activated deglycosylated vitamin D binding protein (dgVDBP). Intravenous application of this dimeric compound in mice revealed non-toxic activity using two different concentrations of VitD_3_ bound to the same protein concentration [21].

In this study, we investigated if this water-soluble VitD-dgVDBP was able to up- or downregulate the formation of reactive oxygen substances (ROS) during phagocytosis on macrophages compared to LPS and dgVDBP alone and how it influenced Th1, Th2, and other lymphatic cells in the expression of several markers of clusters of determination (CD3+, CD4+, CD8+, CD14+, and CD16+) before and after application of VitD-dgVDBP to nPBMC cells. Additionally, we used CD45k, a leucocyte antigen increased in autoimmune diseases and cancer [22], CD85k, a killer cell inhibitor receptor on T cells in B-chronic lymphocytic leukemia, and interleukin-3 receptor CD123 to identify any activation of the dimeric VitD-dgVDBP compound.

## 2. Materials and Methods

### 2.1. Materials

Gc-protein was purchased from Sigma Aldrich (G8764; Vienna, Austria). The production of dgVDBP by the stepwise treatment of purified VDBP protein with immobilized b-galactosidase and sialidase to produce dgVDBP was carried out according to Yamamoto et al. [17]. The immobilized enzymes were removed by centrifugation, and VitD-dgVDBP was established as described elsewhere by incubating equimolar concentrations of VitD_3_ [13]. Goat-anti-human-Gc-IgG (SAB2501100) and rabbit-anti-Goat-IgG-HRP (AP106P) were obtained from Sigma Aldrich (Vienna, Austria). The protein determination of VDBP, dgVDBP, and VitD-dgVDBP was performed with the PierceTM BCA Protein Assay kit (Thermo Fisher Scientific, Vienna, Austria). PBMCs from healthy human Caucasian donors (nPBMCs) and prostate cancer cells were purchased from (SER-PBMC-CUSTOM-F-ZB; BioCat; Heidelberg, Germany).

### 2.2. Determination of Vitamin D Binding Protein (VDBP)

This enzyme immunoassay is a sandwich assay for the quantitative determination of VDBP proteins (Immundiagnostik AG, Bensheim, Germany). The wells of the microtiter plate were coated with polyclonal anti-VDBP antibodies. In the first step of incubation, the VDBP in the pre-diluted samples was bound to the wells coated with polyclonal rabbit antibodies. To remove all unbound substances, a washing step was carried out. In the second step, a polyclonal peroxidase-labeled rabbit anti-VDBP antibody was added. After another washing step, to remove all unbound substances, the solid phase was incubated with a substrate, tetramethylbenzidine. An acidic stopping solution was then added. The color changed to yellow. The intensity of the yellow color was directly proportional to the VDBP concentration in the sample. A dose–response curve of the absorbance unit (optical density, OD at 450 nm) vs. the concentration was generated using the values obtained from the standard. The levels of VDBP present in the samples were determined directly from this curve.

### 2.3. Labeling of VitD-dgVDBP

For the protein labeling of VitD-dgVDBP, the Lightning-Link^®^ PE-Cy7 Tandem Conjugation Kit (Lightning-Link^®^, Innova Bioscience, Cambridge, UK) was used. The labeling of VitD-dgVDBP was performed according to the supplier’s instructions. Aliquots were stained with an OptiClone lymphocyte panel or IO Test 3 for monocytic cells, either with or without each labeled protein. At least 30.000 cells were analyzed per sample using a Cytomics FC 500 MPL cytometer (Beckman Coulter, Villepinte, France).

### 2.4. Macrophage Phagocytosis and Superoxide Anion Radical Production Analysis

The quantitative determination of the phagocytic activity of monocytes in heparinized human whole blood with 100 ng/mL LPS, 400 pg/mL dgVDBP, and 400 pg/mL VitD-dgVDBP was performed with the PhagotestTM (Glycotope Biotechnology, Heidelberg, Germany) according to Greilberger et al. [23]. The phagocytosis test kit contained fluorescein-labeled opsonized Escherichia coli bacteria. Heparinized whole blood was incubated with reagent B (FITC-labeled E. coli bacteria) at 37 °C; a negative control sample remained on ice. The phagocytosis was stopped by placing the samples on ice and adding reagent C (quenching solution). This solution allowed for the discrimination between the attachment and internalization of bacteria quenching the FITC fluorescence of the surface-bound bacteria, leaving the fluorescence of internalized particles unaltered. After two washing steps with reagent A (wash solution), the erythrocytes were then removed by the addition of reagent D (lysing solution). A DNA staining solution (reagent E) was added just prior to the flow cytometric analysis and excluded the aggregation artifacts of the bacteria and cells. The E. coli bacteria were opsonized with immunoglobulin and a complement of pooled sera. The cells were analyzed by flow cytometry in the absence or presence of 1 ng/mL LPS, 400 pg/mL dgVDBP, and 400 pg/mL VitD-dgVDBP using blue-green excitation light (488 nm argon ion laser). PhagoburstTM was used for the determination of leukocyte oxidative burst (Glycotope Biotechnology, Heidelberg, Germany) in the presence of 100 ng/mL LPS, 400 pg/mL dgVDBP, and 400 pg/mL VitD-dgVDBP. The PhagoburstTM kit contains unlabeled opsonized E. coli bacteria (reagent B) as a particulate stimulus, protein kinase C ligand phorbol 12-myristate 13-acetate (PMA, reagent D) as a strong stimulus, chemotactic peptide N-formylMetLeuPhe (fMLP, reagent C) as a weak physiological stimulus, dihydrorhodamine (DHR) 123 (reagent E) as a fluorogenic substrate, and necessary reagents. The heparinized whole blood was incubated with the various stimuli at 37 °C. A sample without stimulus served as a negative background control. Upon stimulation, granulocytes and monocytes produce reactive oxygen metabolites (superoxide anion, hydrogen peroxide, and hypochlorous acid), which destroy bacteria inside the phagosome. The formation of the reactive oxidants during the oxidative burst was monitored by the addition and oxidation of DHR 123. The reaction was stopped by the addition of lysing solution (reagent F), which removed the erythrocytes and resulted in a partial fixation of leukocytes. After one washing step with the wash solution (reagent A), a DNA staining solution (reagent G) was added to exclude the aggregation artifacts of bacteria and cells. The percentage of cells having produced reactive oxygen radicals in the absence or presence of 1 ng/mL LPS, 400 pg/mL dgVDBP, and 400 pg/mL VitD-dgVDBP were then analyzed, as well as their mean fluorescence intensity (enzymatic activity, 488 nm excitation argon ion laser).

### 2.5. Apoptosis Measurements on Malignant Prostate Cell Line in the Absence or Presence of dgVDBP or VitD-dgVDBP

The apoptosis of the human prostatic cancer cell line in the absence or presence of dgVDBP and VitD-dgVDBP was measured at the Research Genetic Cancer Center (Florina, Greece) and cultured in RPMI 1640 containing 10% heat-inactivated fetal bovine serum, 100 units/mL penicillin G, and 100 µg/mL streptomycin at 37 °C in 5% CO_2_.

Malignant prostate cancer cells were cultivated on 24 microtitration plates with (200 ng/mL) VitD-dgVDBP, (200 ng/mL) dgVDBP, or without ingredients in the culture media (RPMI) containing 10% fetal calf serum. After 24 h of incubation, these cell cultures were harvested, and the activities of caspase 3, caspase 9, and cytochrome-c were measured using an oncogene apoptosis kit (Abcam, Szabo-Scandic, Vienna, Austria).

### 2.6. Flow Cytometry of T Cells in the Absence or Presence of VitD-dgVDBP or Labeled VitD-dgVDBP

Human peripheral blood mononuclear cells (PBMCs; ZenBio Inc., Durham, NC, USA) were resuspended and divided into aliquots for a native control and incubation with VitD-dgVDBP for 12 h.

Each section contained three aliquot measurements with (1) CD4/CD8/CD3, (2) CD14/CD16/CD45, and (3) pDC (CD14+CD16+/CD85k(ILT3)-PE/CD123-PC5), and immune phenotyping was carried out using monoclonal antibodies (Beckman Coulter, France). Before measurement, 20 µL of BD Tritest™ CD4/CD8/CD3 (Becton Dickinson and Company BD Biosciences, 2350 Qume Drive, San Jose, CA 95131 USA), BD Tritest™ CD14+/CD16+/CD45k+, and BD Tritest™ CD14-CD16/CD85k+/CD123 reagent were pipetted into the bottoms of the according tubes, incubating the cell/antibody mixture for 30 min at room temperature. VitD-dgVDBP was added to the aliquots using a reference concentration of 40 I.U. cholecalciferol at a 1:1 binding ratio to dgVDBP. The cells were analyzed using a Beckman Coulter Cytomics FC 500 MPL device (Beckman Coulter, France). Aliquot 1 cells were gated to CD3+ cells, and then CD4+, CD8+, and CD4+/CD8+ were measured. Aliquot 2 cells were gated to CD45k+ cells, and then CD14+, CD16+, CD16++, and CD14+/CD16+ were measured. Aliquot 3 cells were gated to CD14+Cd16+ cells, and then CD85k+, CD123+, and CD85k+/CD123+ were measured. After another 12 h, the same measurements were carried out with incubated VitD-dgVDBP dimer.

### 2.7. Statistical Analysis

Group comparisons were made using *t*-tests where appropriate and indicated. Linear regression and exponential regression curves were calculated based on the Pearson regression (SPSS 25, SPSS Inc., Chicago, IL, USA). All values are given as the mean values and standard deviations. Statistical significance was considered to be at *p* < 0.05, with high significance at *p* < 0.01.

## 3. Results

### 3.1. Macrophage Phagocytosis and Superoxide Anion Radical Production Analysis

It was suggested that dgVDBP and VitD-dgVDBP can initiate macrophage activation and radical formation in a different manner; therefore, macrophage activation and superoxide anion radical formation were measured as described in the methods. VitD-dgVDBP showed a significant lower generation of superoxide anion radicals compared to dgVDBP (355,145 ± 82,750 vs. 630,000 ± 54,250 fluorescence counts; *p* < 0.001; *n* = 5) and a significant activation of macrophage activity, twice as high, using the same protein concentration (14,777,500 ± 1,889,500 vs. 6,998,750 ± 216,520 fluorescence counts; *p* < 0.001; *n* = 5). Using the same protein concentration, LPS resulted in a reduced signal in the generation of superoxide anion radicals and macrophage activation compared to dgVDBP and VitD-dgVDBP (432 ± 24.7 and 2866.7 ± 102 fluorescence counts; *p* < 0.001, *n* = 5; Figure 1A). Figure 1B shows the ratio of generated superoxide anion radical to macrophage phagocytosis activation setting the LPS ratio to 1. The lowest ratio was measured with VitD-dgVDBP (0.16 ± 0.02), which was nearly 4-fold lower compared to dgVDBP (0.61 ± 0.04; *p* < 0.001) and more than 6-fold lower compared to LPS (1 ± 0.07; *p* < 0.001), whereas dgVDBP appeared to be slightly lower compared to LPS (*p* < 0.01).

### 3.2. Apoptosis Measurements on Malignant Prostate Cell Line in the Absence or Presence of dgVDBP or VitD-dgVDBP

Stimulated macrophages can conduct different pathways that also involve apoptosis via caspase-activating processes. Using cell media with or without dgVDBP or VitD-dgVDBP, no apoptosis was estimated with dgVDBP compared to the blank (cell media) using caspase-3 activity (2.4 ± 0.1% vs. 2.4 ± 0.1%, *n* = 3), caspase-9 activity (2.2 ± 0.1% vs. 2.2 ± 0.2%, *n* = 3), and cytochrome-c activity (2.3 ± 0.1% vs. 2.3 ± 0.3%, *n* = 3) as presented in Figure 2.

The dimeric compound VitD-dgVDBP showed in Figure 2 significantly high apoptosis compared to the blank or dgVDBP in caspase-3 activity (10 ± 1.4%, *n* = 3; *p* < 0.01), caspase-9 activity (15 ± 2.0%, *n* = 3; *p* < 0.01), and cytochrome-c activity (14 ± 1.4%, *n* = 3; *p* < 0.01).

### 3.3. Flow Cytometry of T Cells in the Absence or Presence of Labeled VitD-dgVDBP

In this study, we investigated for the first time the binding of VitD-dgVDBP to leucocytes. We labeled the dgVDBP of the VitD-dgVDBP and incubated it with all leucocytes. Figure 3 presents the binding of fluorescence-labeled VitD-dgVDBP to cytotoxic cells (CD8+) before (Figure 3A,C) and after the addition of VitD-dgVDBP (Figure 3B,D) to the nPBMCs of healthy persons on suppressor cells.

Figure 4 presents the binding of CD4+ and CD8+ before (Figure 4A) and after the addition of labeled VitD-dgVDBP (Figure 4B) to the nPBMCs of healthy persons.

### 3.4. Flow Cytometry of nPBMCs in the Absence or Presence of VitD-dgVDBP

The purchased nPBMCs were classified with the cluster of determination parameters presented in Table 1.

Table 2 shows the effects of added VitD-dgVDBP on nPBMCs. A significant 1.5-fold increase was estimated after incubation on CD16++/CD16+ (0.4 ± 0.1% vs. 0.6 ± 0.1%; *p* < 0.05; Figure 5), while VitD-dgVDBP significantly affected a nearly 80% decrease in CD14+/CD16+ (3.1 ± 1.7% to 0.7 ± 0.2%; *p* < 0.05; Figure 5). CD45k^+^ showed a significant 13% increase in the presence of VitD-dgVDBP (96.0 ± 6.0%, *p* < 0.05; Figure 5), as did 6-fold with CD85k+ (3.8 ± 3.2% to 24.3 ± 13.2%; *p* < 0.05; Figure 5) and 13.3-fold with CD85k^+^/CD123^+^ (3.5 ± 3.7% to 46.8 ± 8.1%; *p* < 0.001; Figure 5). No differences were seen on the other CDs, namely CD3+, CD4+, CD8+, CD4/CD8, CD4/CD8++, CD16+, CD16++, CD14+, CD14+CD16+, and CD123+. Additionally, we detected nearly the same percentages of CD3, CD4, CD8, and CD14 as described by the customer.

## 4. Discussion

Vitamin D and its metabolites are not only involved in endocrine interactions among the kidney, bone, and parathyroid hormone but also in immunity, blood pressure, muscular function, T-cell regulation, antioxidative defence via glutathione metabolism, autoimmune disease, autism, and cancer defence [24,25,26,27,28,29]. All vitamin D metabolites are mainly bound to vitamin D binding protein (VDBP or Gc), whereas vitamin D_3_ seems to have the highest binding activity compared to calcidiol or calcitriol. During this binding, the specific domain of VDBP, which is absolutely distinct from the domain of the intracellular vitamin D receptor [30], experiences a substantial structural conformation [31].

The deglycosylation of sialic acid and galactose from amino acid threonine at position 436 of the VDBP (GC1 allele) induced a higher binding affinity to cholecalciferol compared to the fully glycosylated protein, accompanied by a further conformational change of the protein structure, which resulted in a higher monoclonal antibody signal against vitamin D [23].

This dimeric form of VitD-dgVDBP is able to increase macrophage phagocytosis much better than the non-VitD-loaded form of dgVDBP. The dgVDBP also increased the generation of superoxide anion radicals, and the VitD-dgVDBP showed significantly lower induction; the ratio of generated radicals to macrophage phagocytosis activity was more than 7-fold lower in VitD-dgVDBP compared to LP, and S 5-fold lower compared to non-bound dgVDBP.

This might be caused by the known fact that VDBP can also transport polyunsaturated fatty acids (PUFAs). These PUFAs are further oxidized during macrophage phagocytosis by lipid peroxidation to several lipid radicals, e.g., lipid peroxides, lipoxy radicals, and their metabolites, malondialdehyde, and hydroxynonenal. Additionally, the binding capacity of VitD to VDBP and dgVDBP decreases with the binding of polyunsaturated fatty acids on domain II [11]. The water-soluble dimeric VitD-dgVDBP compound seems to have high activation of phagocytosis with low ROS production, which might be due to its synergistic effect.

In addition to the renal uptake by the megalin–cubilin pathway, vitamin D uptake into various cells is not clearly understood. Although some activated T cells can express megalin, they lack cubilin. Therefore, micropinocytosis seems to be the most likely process for VDBP uptake. It is also hypothesized that VitD can be preserved by binding to several VDBP isoforms [12]. The intracellular transport of VitD remains unclear. Some researchers argue the “free hormone hypothesis” because VDBP-null mice or humans have low VitD levels with intact calcium levels and normal bone homeostasis, which may also result from VitD bound to albumin. We have reported that VitD-dimeric compounds containing 1.2 IE VitD to dgVDBP (ImmunoD) and 0.6 IE VitD to the same amount of dgVDBP (PolyNac) intravenously applied in mice did not have any change in calcium levels after 5 weeks. These findings support the free hormone hypothesis. Interestingly, however, the lower VitD form (PolyNac) generated higher serum VitD levels than ImmunoD, which significantly increased the serum vitamin D levels compared to the control.

This experiment demonstrates that dgVDBP, and possibly VDBP, does have an inhibiting effect during the intracellular uptake of VitD into cells. This has also been reported in T cells, where VDBP downregulates the conversion to calcitriol by CYP27B1 after sequestering VitD [12]. This regulation of T-cell response to vitamin D levels by VDBP was measured by the increase in serum VitD in VitD-dgVDBP-treated mice. The inhibition was affected by neither actin nor arachidonic acid. Furthermore, the carbonylation of VDBP by oxidative stress is able to detach VitD due to its lower affinity for carbonylated VDBP. The oxidized form of VDBP can be detected in human serum. During the activation of phagocytosis and inflammation, ROS are generated, which modify VDBP or dgVDBP to carbonylated and degenerated forms. This destabilizes the binding of vitamin D to VDBP and leads to the loss of VDBP function.

We demonstrated for the first time that binding to T cells takes place with water-soluble and fluorescence-labelled VitD-dgVDBP to CD8-expressing cells, which includes all suppressor and cytotoxic cells, including Treg and natural killer and dendritic cells. This is accompanied by a massive increase in CD8 expression after 12 h of incubation with VitD-dgVDBP compared to non-incubated nPBMCs. A nearly equal binding was found in the CD4/CD8 ratio. Interestingly, CD4/CD8 expression was not affected on lymphocytes in the nPBMCs, not even after 12 h of incubation with VitD-dgVDBP. Monocytes can mature into macrophages or dendritic cells (DC). The latter pathway has achieved much more attention. Recently, the development of DC with the preferential induction of TH2 cells from CD16+ monocytes has been demonstrated [32,33].

Randolph et al. [34] showed that CD14+CD16+ monocytes do preferentially migrate from the bloodstream and develop into DCs with superior stimulatory capacity. Unfortunately, in many of these functional studies, the cells were isolated by positive selection with mAb against cell surface molecules, such as CD16. In our study, we incubated aliquots of nPBMCs in the absence or presence of the water-soluble VitD-dgVDBP dimer complex and measured parallel samples under the same conditions to eliminate a suspected bias due to the mAb influence in previous reports. In this study, we present for the first time that labelled VitD-dgVDBP bound and increased one of the two major types of activated monocytes, namely CD16^++^, after 12 h of incubation. Interestingly, the expression of CD14 was not affected by VitD-dgVDBP in this manner.

LPS is known to enhance the release of ROS, and CD14+ is the key player in LPS-induced ROS production [35].

The independency of this pathway suggests that VitD-dgVDBP expresses more CD16++ than CD14+ selectively. This corresponds with the lower ROS ratio during macrophage activation through VitD-dgVDBP compared to LPS. With the advantage of using water-soluble vitamin D in the form of a newly developed VitD-dgVDBP, we incubated it directly with PBMCs from pooled donors to investigate any influence on the expression of CDs. The expressions of CD3^+^, CD4^+^, CD8^+^, CD14^+^, and CD16^+^ were not affected by VitD-dgVDBP, while CD16++/CD16+, CD45k^+^, CD85k^+^, and the ratio of CD85k^+^/CD123^+^ increased. The CD14+/CD16++ ratio decreased, confirming the independent pathway of VitD-dgVDBP because of a higher expression in favor of CD16++.

CD45k^+^, a protein tyrosine phosphatase receptor type C (PTPRC), is a transmembrane glycoprotein expressed on almost all hematopoietic cells except for mature erythrocytes and is an essential regulator of T and B cell antigen receptor-mediated activation. It was described to possess positive or negative regulation in the immune function of immune cells [36]. CD45k^+^ may play an essential role in innate immune defence: CD45k^+^ affects cytokine, NK receptors, and Toll-like receptors. It also has an effect on the interaction between T cells and macrophages using its macrophage galactose-type lectin, which binds to CD45k^+^ *N*-acetlygalactosamine. This CD45 *N*-acetylgalactosamine has the potential to lower T-cell proliferation by the generation of proinflammatory cytokines, which mostly results in T-cell apoptosis [37]. VitD-dgVDBP contains the protein *N*-acetylgalactosamine, which may have counteracted CD45k^+^ in its regulatory processes on the immune system after incubation on the PBMCs of our healthy population.

CD85k^+^ has an inhibitory regulation effect on NK and T-cell-mediated cytokine production. We obtained the same percentage of CD85 cells in a healthy population compared to another report [38]. A higher expression of CD85 was estimated in the healthy population after incubation with VitD-dgVDBP compared to the non-incubated, but it was still within the range of the reported healthy donors.

While we found no difference in CD123+ between VitD-dgVDBP-treated and non-treated lymphocytes isolated from the healthy population, the ratio of CD85k^+^/CD123^+^ was 10-fold higher in the treated form, which seems to have affected dendritic cells.

CD123 is found on pluripotent progenitor cells, which induces tyrosine phosphorylation within the cell and promotes proliferation and differentiation within the hematopoietic cell lines. The ratio of CD85/CD123 is involved in the characterization of human dendritic cells [39,40].

We estimated for the first time that the dimeric compound VitD-dgVDBP induces apoptosis, whereas dgVDBP does not. While DBP-maf (dgVDBP) was demonstrated to inhibit proliferation, migration, and uPAR expression of prostate cancer cells, no inhibitory effect of dgVDBP was estimated due to apoptosis directly [41]. Thyer et al. also demonstrated the macrophage-stimulated apoptosis method of the single compound dgVDBP (GcMaf) [42] with higher ROS activity.

We confirmed these findings also with the dimeric and water-soluble compound VitD-dgVDBP, partially because the dimeric compound is able to induce apoptosis directly measured by caspase-3, caspase-9, and cytochrome-c activity on prostatic cancer cells, as well as over the macrophage-stimulating phagocytose of several cytotoxic cells, including dendritic cells, natural killer cells, and macrophages together.

## 5. Conclusions

VitD-dgVDBP seems to have immune modulatory effects, binding directly to cytostatic cells and activating the CD16 pathway more but not the CD14 pathway, which is directed by a lower radical activity mechanism. Additionally, VitD-dgVDBP is able to induce apoptosis directly on cancer cells compared to the single component dgVDBP (GcMaf). Further studies on PBMCs and clinical trials are needed for a more in-depth investigation into this newly developed soluble vitamin D form.

## Figures and Tables

**Figure 1 biomedicines-10-01785-f001:**
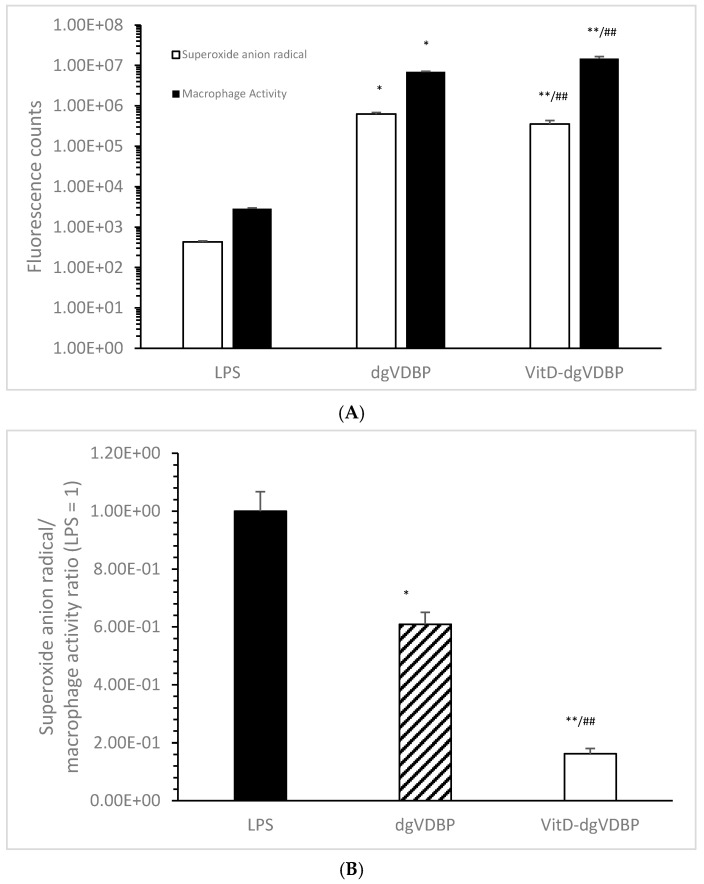
Superoxide anion radical generation (white bars), macrophage activation (black bars; (**A**)), and the ratio of superoxide anion radical/macrophage activation (**B**) of LPS, dgVDBP, and VitD-dgVDBP setting LPS ratio to 1. * *p* < 0.01: significant difference between LPS and dgVDBP; ** *p* < 0.001: significant difference between LPS and VitD-dgVDBP; ^##^ *p* < 0.001: significant difference between dgVDBP and VitD-dgVDBP. 1.00E+08 = 1 × 10^8^.

**Figure 2 biomedicines-10-01785-f002:**
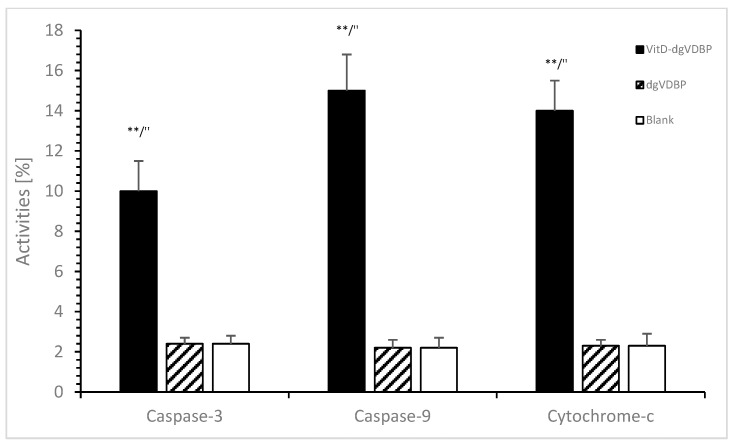
Apoptosis of prostate cancer cell line in the presence or absence of dgVDBP or VitD-dgVDBP using the detection of caspase-3, caspase-9, and cytochrome-c. ** *p* < 0.01: significant difference between VitD-dgVDBP and cell media; ″ *p* < 0.01: significant difference between VitD-dgVDBP and dgVDBP.

**Figure 3 biomedicines-10-01785-f003:**
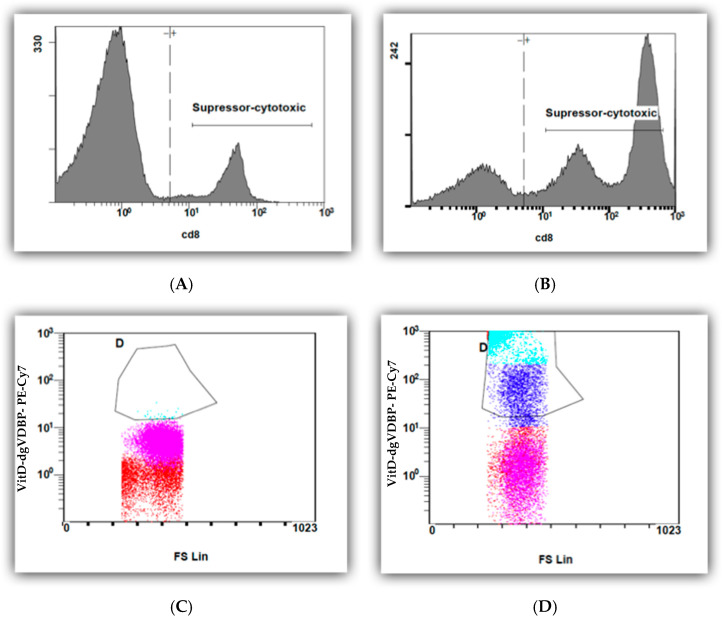
Flow cytometry measurements of CD8 cells of nPBMCs of healthy person before (**A**,**C**) and after (**B**,**D**) application of fluorescence-labeled VitD-dgVDBP.

**Figure 4 biomedicines-10-01785-f004:**
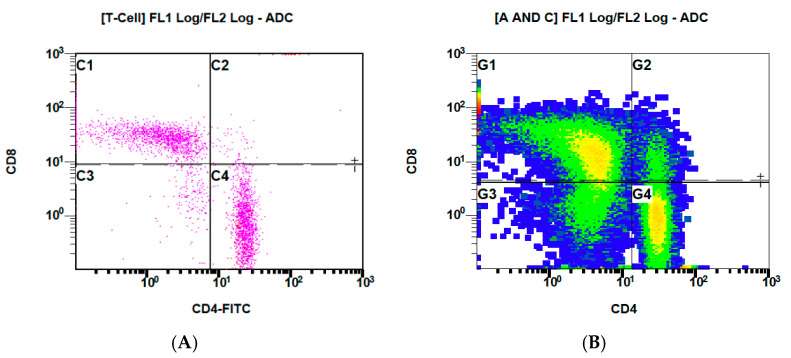
Flow cytometry measurements of CD4/8 expressed on nPBMCs of healthy persons before (**A**) and after (**B**) application of labeled VitD-dgVDBP.

**Figure 5 biomedicines-10-01785-f005:**
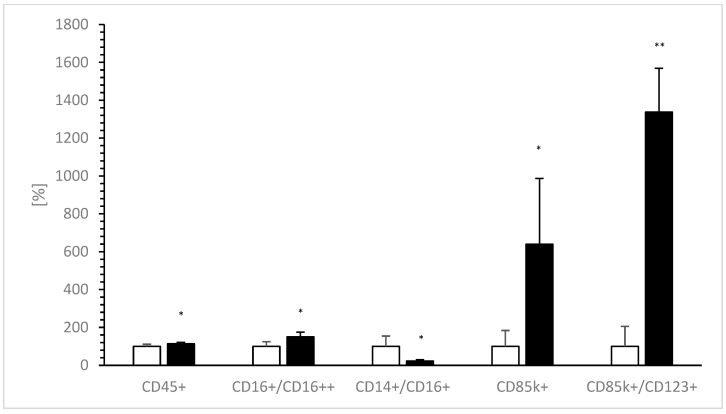
Expressions of CD14+/CD16+, CD16++/CD16+, CD45k^+^, CD85k^+^, and ratio of CD85k^+^/CD123^+^ before (0 h of incubation, white bars) and 12 h after incubation (black bars) of nPBMCs with VitD-dgVDBP. Each CD without application of VitD-dgVDBP was set to 100% and compared to VitD-dgVDBP incubated CDs. * *p* < 0.05: significant difference between 0 h and 12 h of incubation with VitD-dgVDBP. ** *p* < 0.001: significant difference between 0 h and 12 h of incubation with VitD-dgVDBP.

**Table 1 biomedicines-10-01785-t001:** Clusters of differentiation (CDs) of purchased nPBMCs according to the data sheet of the customer.

CD3 (T Cells) %	CD4 (TH) %	CD8 (TC) %	CD19 (B Cells) %	CD14 (Monocyte) %	CD56/CD16 (NK Cells) %	PROPIDIUM IODIDE % Viability	HLA Type (ABC Type)	HLA-DR% (CLASS II)
56.4	43.2	14.6	9.3	14.6	1.7	87	HLA-A2 NEG, B27+	24.7

**Table 2 biomedicines-10-01785-t002:** Clusters of differentiation (CDs) of nPBMCs (*n* = 10) before incubation (0 h) with VitD-dgVDBP and after 12 h of incubation with VitD-dgVDBP.

	0 h Incubation		12 h Incubation with VitD-dgVDBP		
	Mean (% Lymph)	SD (% Lymph)	Mean (% Lymph)	SD (% Lymph)	*p*
CD3+	61.9	5.7	68.5	4.7	n.s.
CD4+	55.7	3.7	60.8	3.3	n.s.
CD8+	15.6	8.2	13.6	4.6	n.s.
CD4/CD8	2.1	1.1	2.0	1.2	n.s.
CD4/CD8++	0.5	0.3	0.6	0.3	n.s.
CD45+	84.7	9.5	96.0	6.0	*p* < 0.05
CD16+	10.4	4.0	8.4	4.5	n.s.
CD16++	4.5	2.9	6.2	3.4	n.s.
CD16+/CD16++	0.4	0.1	0.6	0.1	*p* < 0.05
CD14+	18.2	6.0	17.6	6.1	n.s.
CD14+/CD16+	3.1	1.7	0.7	0.2	*p* < 0.05
CD14+CD16+	14.6	4.7	20.5	6.7	n.s.
CD85k+	3.8	3.2	24.3	13.2	*p* < 0.05
CD123+	52.4	11.2	61.9	7.3	n.s.
CD85k+/CD123+	3.5	3.7	46.8	8.1	*p* < 0.001

## Data Availability

The data presented in this study are available in the article.

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
