# Peer review of "Vitamin D-Dimer: A Possible Biomolecule Modulator in Cytotoxic and Phagocytosis Processes?"

_biomedicines, 2022, doi:10.3390/biomedicines10081785_

Round 1

Reviewer 1 Report

Abstract section: please improve the introduction part as currently, it is not well written and can be misleading to the reader.

Author Response

Dear reviewer

thank you for your comments. In the attached file we have made all corrections as possible.

Reviewer 2 Report

In the paper entitled "Vitamin D -Dimer: A possible biomolecule modulator in cytotoxic and phagocytosis processes?" the authors report on the role played by VitD-dg-VDBP, a deglycosylated vitamin D binding protein, in inducing apoptosis directly on cancel cells.

Despite the interest of the topic, the research results reported in the paper are very difficult to analyze and manage. The result section is very poor: the experiments that have been carried out to test the activity of VitD-dg-VDBP have not been discussed and the date have been reported in graphs that sometimes are not so accurate.

In the following the main critical points that should be addressed:

1) The abstract section is too long: it should not exceed the 200 words and more than 500 words are reported. Moreover, the abstract should be structured in the following sections: background, methods, results and conclusion that are not so evident.

2) The structures of the the different kind of vitamin D described in the paper should be shown in a figure just to make the comparison more easy.

3) The caption of Figure 1 is not clear. The graph A shows two different sets of data reported with white and black bars, but there is no mention on what the white and black bars are referred to.

4) In the graph reported in Figure 2, the title of the y-axis should be explicited.

5) All the tests performed and reported in the Figures 1-4 should be introduced in a better way. Please expand this section.

6) In the Table 2, some date have been subjected to statistical analysis, but it has not been indicated the reference data.

7) The graphs in Figure 5 report the same data of Table 2. I suggest to move these graphs in the supporting section.

8) An extensive english editing is mandatory.

Due to the many criticisms reported above, the paper is not suitable for publication in Biomedicines in the present form.

Author Response

(The authors gave the same response as above.)

Reviewer 3 Report

Topic is suitable for the Biomedicine, but its quality muss be significantly improved.

One abbreviation is used for two different terms. Line 23-24 Water soluble Vitamin D bound to deglycosylated Vitamin D 24 Binding Protein (VitD-dgVDBP) Line 33 Deglycosylated Vitamin D Binding Protein (VitD-dgVDBP)

Line 46-47 fluorescence value 19777500 +/- 1889500 is written in a misleading way. In these cases, appropriate rounding should be done.

Line 136 VitD-dgVDBP most probably is not dimeric compounds, but complex of the vitamin D with protein dimer.

Line 164 VitD dimer, it was mentioned VidD-dgVDBP?

2. materials and method source, nor preparation of VidD-dgVDBP is not writen

Line 181-182 different format of text

Figure 1 A black, or white is not defined.

Figure 2 % of what?

Figure 3 label of y axis is not defined.

Table 2 Significantly difference can be written bold

Author Response

(The authors gave the same response as above.)

Round 2

Reviewer 2 Report

I appreciate the extensive work made by the authors in revising the paper. All my suggestions have been addressed and the paper looks fine now.

I have only minor revisions to report:

In Figure 1B the authors report on the ratio between the fluorescence counts from superoxide anion radicals and macrophage activity. The values have been taken from the graph A, I suppose. In this case the ratio should be <1.

The graphs in Figure 5A-E can be recombined trying to show the data in a more compact way. Probably, the authors can provide a single graph with the effect of the incubation of nPBMCs with VitD-dgVDBP on the expression of the selected monocytes.

Overall, I can reconsider the submission of the paper in Biomedicines after minor revisions 

Author Response

Dear reviewer

we changed some minor points in Figure1B and Figure 5 for rev.2.

As there are no additional comments of you we do thank you for your commends which improofs our presentation of our data in this manuscript!

Best regards

Herwig and Greilberger

Reviewer 3 Report

I have no objection.

Author Response

Dear editor,

we took a native speaker in our first revision and due to the fact that you have no objection we changed the manuscript to the comments of rev. 2.

Thank you for your comments

Herwig and Greilberger